# The Small Ras Superfamily GTPase Rho4 of the Maize Anthracnose Fungus *Colletotrichum graminicola* Is Required for β-1,3-glucan Synthesis, Cell Wall Integrity, and Full Virulence

**DOI:** 10.3390/jof8100997

**Published:** 2022-09-22

**Authors:** Ely Oliveira-Garcia, Lala Aliyeva-Schnorr, Alan De Oliveira Silva, Seif El Din Ghanem, Kathrin Thor, Edgar Peiter, Holger B. Deising

**Affiliations:** Institute of Agricultural and Nutritional Sciences, Faculty of Natural Sciences III, Martin Luther University Halle-Wittenberg, Betty-Heimann-Str. 3, D-06120 Halle, Saale, Germany

**Keywords:** appressoria, β-1,3-glucan synthesis, *Colletotrichum graminicola*, fungal cell walls, hyphopodia, penetration defect, Rho GTPases, virulence factor, *Zea mays*

## Abstract

Small Ras superfamily GTPases are highly conserved regulatory factors of fungal cell wall biosynthesis and morphogenesis. Previous experiments have shown that the Rho4-like protein of the maize anthracnose fungus *Colletotrichum graminicola*, formerly erroneously annotated as a Rho1 protein, physically interacts with the β-1,3-glucan synthase Gls1 (Lange et al., 2014; Curr. Genet. 60:343–350). Here, we show that Rho4 is required for β-1,3-glucan synthesis. Accordingly, Δ*rho4* strains formed distorted vegetative hyphae with swellings, and exhibited strongly reduced rates of hyphal growth and defects in asexual sporulation. Moreover, on host cuticles, conidia of Δ*rho4* strains formed long hyphae with hyphopodia, rather than short germ tubes with appressoria. Hyphopodia of Δ*rho4* strains exhibited penetration defects and often germinated laterally, indicative of cell wall weaknesses. In planta differentiated infection hyphae of Δ*rho4* strains were fringy, and anthracnose disease symptoms caused by these strains on intact and wounded maize leaf segments were significantly weaker than those caused by the WT strain. A retarded disease symptom development was confirmed by qPCR analyses. Collectively, we identified the Ras GTPase Rho4 as a new virulence factor of *C. graminicola*.

## 1. Introduction

The cell wall determines the shape of the fungal cell, protects it from environmental stresses, and, in pathogenic fungi, represents the interface with the host. The latter aspect is of particular importance, as major cell wall polymers such as chitin or branched β-glucans are on the one hand required for cell wall function, but on the other represent conserved pathogen-associated molecular patterns (PAMPs). PAMPS, upon perception by immune receptor complexes (pattern recognition receptors; PRRs), activate a broad range of defense responses in plants [1,2,3,4,5]. As a result of these conflicting functions, the formation as well as the surface exposure of cell wall polymers is highly dynamic in infection structures of many fungal pathogens and is often associated with infection-related morphogenesis [6].

Depending on their mode of invasion and lifestyle, fungal pathogens form infection structures differing in complexity and function [7,8]. The formation of infection cells called appressoria on the plant cuticle initiates host invasion [9,10,11,12]. Interestingly, *Colletotrichum*, *Magnaporthe*, and *Phyllosticta* species form melanized appressoria, which generate an enormous turgor pressure of up to 80 bar [13]. In order to invade the first host epidermal cell, the turgor pressure is translated into force, which is exerted onto the plant surface on an appressorial basis [11,14,15]. Plausibly, as branched β-glucan, as well as chitin, represent key structural cell wall polymers, the formation of these polymers is indispensable for appressorium function. As indicated by RNA interference (RNAi) experiments in the maize anthracnose fungus *Colletotrichum graminicola*, mutants with reduced transcript abundances of the β-glucan synthesis genes *GLS1*, *KRE5,* or *KRE6* showed reduced contents of branched β-glucan in cell walls, and appressoria exploded on the host surface due to reduced cell wall rigidity [5,16]. However, although indispensable in appressoria, exposure to β-glucans on the surface of invading biotrophic hyphae of this fungus is detrimental, as these hyphae are surrounded by the host plasma membrane harboring PRRs recognizing conserved cell wall components, including β-glucan fragments [1,2,3,17]. Due to high PRR densities [17], one may expect that PAMP exposure on the surface of invading hyphae is rigorously avoided to circumvent defense induction [6]. Indeed, in biotrophic hyphae of *C. graminicola*, branched β-glucan formation is strongly reduced, as indicated by labeling with fluorescence-tagged aniline blue and a YFP-tagged β-1,6-glucan-binding protein [5,16]. The requirement for the tight regulation of β-glucan formation was demonstrated by overexpressing *GLS1*, *KRE5*, and *KRE6* in biotrophic hyphae, which caused strong PAMP-triggered immune (PTI) responses in maize leaves and failure of infection [5,16]. Thus, genes required for β-glucan synthesis are infection stage-specifically regulated, and identifying corresponding regulatory factors is indispensable for understanding cell wall biogenesis and infection biology, and may lead to the discovery of novel targets addressed by antifungal chemistries in plant protection.

Ras-homologous Rho GTPases are highly conserved determinants of several aspects of fungal morphogenesis, including cell wall biosynthesis. Rho GTPases are moved from an inactive GDP-bound to an active GTP-bound state by guanine nucleotide exchange factors, which thus act as key regulators of Rho-type GTPases [18]. Rho GTPases function as molecular switches and execute distinct regulatory roles, e.g., in cell wall β-glucan formation, cell wall integrity, cell polarity, and development [19,20,21,22] (references therein). The genomes of the unicellular model yeasts *Saccharomyces cerevisiae* and *Schizosaccharomyces pombe* harbor six Rho GTPase genes each, designated as *rho1* to *rho5* and *cdc42*, with partially overlapping functions. All Rho proteins have a *C*-terminal prenylation site, and prenylation is required for their plasma membrane association and activity. In both yeasts, *rho1* and *cdc42* are essential genes [19,21].

In *S. cerevisiae*, Rho1p co-purified with β-1,3-glucan synthase and associated with the catalytic subunit Fks1p in vivo, indicating that both proteins are members of the β-1,3-glucan synthase complex. Moreover, the thermo-labile glucan synthase activity of a temperature-sensitive Rho1p mutant was restored by the addition of recombinant Rho1p, and glucan synthase from mutants expressing constitutively active Rho1p did not require exogenous guanosine triphosphate (GTP) for activity. Collectively, these data show that the Rho1 protein represents a regulatory subunit of the β-1,3-glucan synthase complex [23].

Not only in yeast but also in the filamentous fungus *Aspergillus fumigatus*, an opportunistic pathogen of mammals, is Rho1p a member of the β-1,3-glucan synthase complex, as suggested by co-precipitation of Rho1p and the β-1,3-glucan synthase protein Fks1p in product entrapment experiments [24]. As Rho1p of *A. fumigatus* is indispensable, strains with conditional and constitutive expressions of *RHO1* were constructed. Reduced *RHO1* expression led to a reduction in β-1,3-glucan contents in cell walls, and the attenuation of virulence in a model host, i.e., the greater wax moth *Galleria mellonella* [25]. Moreover, mutants of the dimorphic mammalian pathogen *Candida albicans* conditionally expressing Rho1p under the control of the glucose-repressible phosphoenolpyruvate carboxykinase promoter showed lysis and cell death under non-permissive conditions. In a mouse model of systemic candidiasis, serum glucose concentrations were sufficient to suppress *RHO1* expression in these strains, resulting in failure of kidney colonization and establishment of disease [26]. Furthermore, in the dimorphic fungus and cotton pathogen *Ashbia gossypii*, homokaryotic Δ*rho1* mycelia were obtained, but developing micro-colonies of these mutants exhibited high rates of cell lysis and death of the colonies even in the presence of osmolytes [27]. Finally, in the wheat head blight fungus *Fusarium graminearum*, attempts to delete the *RHO1* ortholog were unsuccessful, suggesting that Rho1 is indispensable in this plant pathogen as well [28].

Taken together, the above-mentioned studies indicate an important role of Rho1 GTPases in cell wall β-1,3-glucan biogenesis, with the roles of other Rho proteins poorly studied in pathogenic fungi. Here we show that in the maize pathogen *C. graminicola,* a Rho4-like protein is required for β-1,3-glucan synthesis, cell wall integrity, and full virulence.

## 2. Materials and Methods

### 2.1. Fungal Strains, Culture Conditions, Hyphopodium Differentiation, and Virulence Assays

The wild-type (WT) strain M2 (syn. M1.001; https://mycocosm.jgi.doe.gov/Colgr1/Colgr1.home.html; accessed on 19 September 2022) of *C. graminicola* (Ces.) G.W. Wilson, Δ*rho4,* and ectopic strains generated in this study were cultivated on oatmeal agar (OMA) or potato dextrose agar (PDA) at 23 °C under continuous fluorescent light (Climas Control CIR, UniEquip, Martinsried, Germany) [29]. For osmotic support, media were supplemented with 0.15 M KCl or 1 M sorbitol [5]. In cell wall challenge experiments, 0.1 μg/mL Nystatin (Holsten Pharma GmbH, Frankfurt, Germany); 0.1 or 0.2 μg/mL Caspofungin (Cancidas; MSD Sharp & Dome, Haar, Germany) were included in PDA (Difco Laboratories, Sparks, MD, USA) containing 150 mM KCl.

To quantify conidiation rates, fungal strains were grown in Petri dishes (Ø 9 cm) containing OMA with 150 mM KCl. After 14 days, cultures were washed on a rotary shaker with 10 mL aqueous 0.01% (*v*/*v*) Tween 20 for 10 min, and conidia were counted in a Thoma chamber.

To induce infection structure differentiation, conidial suspensions (10^5^/mL 0.02% [*w*/*v*] Tween 20) were inoculated onto the adaxial epidermal mid-rib of 12-day-old *Zea mays* cv. Badischer Landmais (Spinne Walzenmühle, Oelde, Germany) leaves or onto onion (*Allium cepa* cv. Shakespeare) epidermis and incubated at 23 °C in darkness [5]. Virulence assays were performed by inoculating non-wounded or wounded leaf segments of 12-day-old maize (*Z. mays* cv. Badischer Landmais) plants with sterile 10 μL droplets containing 10^6^ conidia/mL 0.02% (*w*/*v*) Tween 20. Mock inoculation was performed with 10 μL droplets of 0.02% (*w*/*v*) Tween 20 without fungal conidia. Leaves were incubated at 23 °C in darkness in moist chambers, and disease symptoms were photographed four days post-inoculation (dpi).

Fungal DNA in infected leaves was quantified as described [30]. Briefly, quantitative PCR (qPCR) was performed using primers Cg_ITS2-F1.1 and Cg_ITS2-R1 targeting the ITS2 region of the rDNA cluster with high specificity and sensitivity, due to the presence of a predicted number of 60 rDNA repeats (http://www.broadinstitute.org/annotation/genome/colletotrichum_group/GenomeStats.html; accessed on 15. May 2022). Sequences of these and all other PCR primers mentioned in this paper are listed in Appendix A.

All experiments were performed in triplicate with three technical repeats.

### 2.2. Genome Mining, Phylogeny, and Identification of Conserved Domains

The phylogenetic tree was constructed according to Dautt–Castro et al. (2021), using protein sequences of the Joint Genome Institute (JGI) Mycocosm database (https://jgi.doe.gov/; accessed on 19 September 2022 [31]), as identified by a search conducted using the PFAM Term “PF00071” which refers to the “Ras Family” [32]. The selection and classification of each protein were based on the provided annotations as well as a confirmatory blast against Rho1p through Rho4p of the *S. cerevisiae* reference genomes (accessions: QHB12411, QHB11287.1, KZV10495.1, QHB10053.1). The resulting sequences were compiled into a FASTA file and aligned via MEGA7 using the MUSCLE algorithm [33,34]. The aligned sequences were imported into an R script to determine the optimal evolutionary model based on Akaike’s Information Criterion (AIC) score [35]. After selecting the optimal model, the conversion to nexus via MEGA7 and import into MrBayes for the construction of phylogenetic trees were performed. MrBayes utilizes Bayesian inference using the Markov Chain Monte Carlo method to propagate model parameters onto the inputted sequence, generating a phylogenetic tree [36]. The calculation was terminated when the average standard deviation of split frequencies reached a value of 0.01. The generated data were visualized using FigTree to create the phylogenetic tree with branch probabilities.

### 2.3. Bioinformatics

In order to visualize differences in the spatial arrangement of amino acid sequences within protein domains, i.e., G-Boxes 1–5 of Rho1–4, comparisons were made between three species, i.e., *C. graminicola*, *A. niger,* and *S. cerevisiae*. Conserved protein domains were identified using the conserved domain database CDD/SPARCLE [37]. The G1 box shows the amino acid sequence GXXXXGK[T/S]; the G2 box identified as a threonine residue overlaps with the Switch I region PT[I/V]FE[N/R/K]Y. The G3 Box is DTAG, the G4 box [N/T]KXD, and the G5 box is [C/S]A[K/L/T]. Finally, the *C*-terminal prenylation domain (PD) was visualized, which occurs at the end of the protein and is characterized by the conserved CaaX motif, representing any aliphatic, and X representing amino acid residue.

The sequences were imported and annotated into jalview [38] to visualize the conservedness of these domains.

### 2.4. Targeted Deletion of C. graminicola RHO4

For the targeted deletion of the 1081 bp *C. graminicola RHO4* gene, the *NatR* cassette was PCR-amplified from pNR1 [39], using primers Nourse1pNR1-fw and Nourse2pNR1-rv. The 1009 bp 5′- and the 1003 bp 3′-flanking regions of the *RHO4* gene were amplified from genomic DNA using primers CgPRho1-fw, CgP1Rho4_5’-flank-rv, and CgTRho4_3’-flank-fw and CgTRho4-rv, respectively. 5′-flank, *NatR*, and 3′-flank were fused by double-joint-PCR [40], and nested primers CgPRho4nest-fw and CgTRho4nest-rv were used to amplify the 4210 bp deletion construct, which was transformed into conidial protoplasts as described [29]. Successful deletion of *RHO4* was confirmed by Southern hybridization [5].

### 2.5. DNA Isolation and Genomic Southern Blot Analyses

DNA isolation was done as described [41]. Southern blot analyses were performed with 10 μg *Xho*I-digested DNA [5]. A 500-bp probe corresponding to the 5′-flank was generated using the primers CgRhoIprobe-Fw and CgRhoIprobe-Rv. Hybridization and probe detection followed the recommended protocol (Roche Diagnostics, Mannheim, Germany). The membrane was exposed to Hyperfilm ECL X-ray film (Amersham Pharmacia Biotech, Piscataway, NJ, USA).

### 2.6. Microscopy and Ratio Imaging

Bright-field, differential interference contrast (DIC) microscopy, and fluorescence microscopy were performed with a Nikon Eclipse 600 microscope (Nikon, Düsseldorf, Germany). Digital images were taken with a Nikon microscope camera DS-Ri2 (Nikon). Image processing was done with NIS-Elements imaging *s**oftware* (Nikon). To detect chitin in the fungal cell wall, a Calcofluor White Stain (Fluka Calcofluor White-M2R 1 g/L, Sigma-Aldrich) was applied to hyphae grown on PDA containing 0.15 M KCl. Sterilized coverslips were placed on the surface of the freshly inoculated agar plate to allow the newly grown hyphae to adhere to the glass. Hyphae of the WT strain had covered the coverslip at 3 dpi, and hyphae of the Δ*rho4* mutant at 8 dpi. A 10 μL droplet of Calcofluor White Stain was incubated with the specimen for 10 min at room temperature in the dark. Fluorescence microscopy was performed using a 40× Plan Fluor or a 60× Plan Apo ʎ lens at an excitation wavelength of 350 nm and a laser light transmission of 25% (ND4 in, ND8 out) (Nikon, Düsseldorf, Germany).

For ratio imaging, an AxioObserver Z1 inverted microscope (Carl Zeiss, Oberkochen, Germany) equipped with a Plan Apochromat 63×/1.40 oil immersion objective and an AxioCam MRm camera was used. Epi-illumination analyses employed the filter set 49 for Aniline Blue fluorochrome and the filter set 38HE for Alexa Fluor 647-Wheat Germ Agglutinin conjugate, respectively. Image acquisition and analyses were performed by using AxioVision 4.8.2 software with the physiology module (all from Carl Zeiss).

β-1,3-glucan was stained with Aniline Blue Fluorochrome (Biosupplies Australia, Parkville Victoria, Australia) as described [5]. For chitin staining, the Alexa Fluor 647-Wheat Germ Agglutinin conjugate (Thermo Fisher Scientific, Waltham, MA, USA) was used as suggested by the manufacturer.

### 2.7. Statistical Analyses

Calculation and statistical analysis of differences between groups were done using the one-way ANOVA test, followed by a Tukey–HSD test at the alpha degree (*p* < 0.05) (Microsoft Excel 2016).

### 2.8. Protein Accession Numbers

Accession numbers of Rho proteins used in this study are available at the JGI Organism Database (accessed on 20 September 2022):

Rho1 protein IDs: *Colletotrichum graminicola*, 2624; *Aspergillus niger* 1147694; *Schizosaccharomyces pombe*, 1778; *Magnaporthe oryzae*, 4593; *Neurospora crassa*, 3503; *Saccharomyces cerevisiae* M3707, 32441. Rho2 protein IDs: *C. graminicola*, 4280; *A. niger*, 1091425; *S. pombe*, 1859; *M. oryzae*, 567; *N. crassa*, 6295; *S. cerevisiae*, 34581. Rho3 protein IDs: *C. graminicola*, 6038; *A. niger*, 1128039; *S. pombe*, 419; *M. oryzae*, 11814; *N. crassa*, 2127; *S. cerevisiae*, 36204. Rho4 protein IDs: *C. graminicola*, 9388; *A. niger*, 1131105; *S. pombe*, 1306; *M. oryzae*, 11290; *N. crassa*, 2850; *S. cerevisiae*, 35222, *Aspergillus fumigatus*, A1163, 105180; *Aspergillus nidulans*, 9493; *Penicillium chrysogenum*, 67435; *Cladosporium fulvum*, 183704; *Alternaria alternate*, 94425; *Botrytis cinerea*, 4991; *Blumeria graminis* f. sp. *hordei*, 611727; *Candida albicans*, 59173; *Arthrobotrys oligospora*, 673; *Cladonia grayi*, 8093; *Trinosporium guianense*, 49956. Ran protein ID of *S. cerevisiae*, 35714.

## 3. Results

### 3.1. The Genome of C. graminicola Harbors a Single-Copy Gene Encoding a Rho4-like Protein

The annotated genome of *C. graminicola* (https://mycocosm.jgi.doe.gov/Colgr1/Colgr1.home.html; accessed on 19 September 2022) [42] harbors four genes encoding Rho family proteins. The sizes and amino acid sequences of the derived proteins, along with the organization of the G-boxes (G1–G5) and prenylation domains, allowed the four Rho proteins to be identified as members of the *C. graminicola* Rho-subfamilies Rho1–Rho4 (Figure 1). The single-copy 1081 bp *RHO4* gene harbors two introns (nucleotides 373–450 and 655–793) and codes for a 287 amino acid protein designated as Rho4. As compared with Rho1, Rho2, and Rho3 proteins, Rho4 proteins, including Rho4 of *C. graminicola*, have an extended N-terminus in front of the first G-box, clearly discriminating this Rho-subfamily from all others (Figure 1A). Like all other Rho proteins, Rho4 of *C. graminicola* has five conserved G-box domains involved in GTP binding and/or hydrolysis (Figure 1B, G1–G5) (amino acids 76–83; 100–106; 125–128; 183–186; 225–227), as well as a C-terminal CAAX prenylation site (amino acids 284–287), where C indicates a cysteine; A, any aliphatic amino acid; and X, any amino acid. The comparison of the amino acid sequences of the G-boxes of the model filamentous ascomycetes *Magnaporthe oryzae*, *N. crassa*, and *Aspergillus niger*, as well as of the yeasts *S. cerevisiae* and *S. pombe*, with those of *C. graminicola* confirmed the conservation of the domains. The phylogenetic tree of Rho GTPases designated distinct branches to the four subfamilies, with Rho4 of *C. graminicola* residing in close vicinity to Rho4 proteins of closely related fungi such as *N. crassa* and *M. oryzae* (Figure 1C). The amino acid similarities of *C. graminicola* Rho4 with the Rho4 proteins of *S. cerevisiae* and *N. crassa* as the most distantly and most closely related proteins were 32.6 and 70.7%, respectively. In conclusion, the amino acid and the domain structure similarities confirm that the gene *RHO4* of *C. graminicola* (GLRG_10328; [42]) encodes a Rho4-like GTPase (NCBI accession number XP_008099204.1).

### 3.2. RHO4-Deficient Mutants Exhibit Phenotypic Similarities with GLS1-RNAi Strains of C. graminicola

To functionally characterize *RHO4* of *C. graminicola*, we generated independent Δ*rho4* deletion strains by homologous integration of a 4057 bp deletion cassette harboring the 2164 bp nourseothricin acetyltransferase gene (*Nat*-1) from *Streptomyces noursei* [39] (Appendix A). The construct was transformed into conidial protoplasts of the *C. graminicola* WT strain M2 (syn. M1.001). Of several independent deletion mutants harboring a single-copy integration of the deletion construct, as shown by Southern blot analyses (Appendix A), three were randomly chosen and, together with the WT strain and a strain carrying a single ectopically integrated copy of the deletion construct (further on called ectopic strain), used for further studies. As expected, in the selected mutants, *RHO4* transcripts were not detected by RT-PCR (Appendix A).

On oatmeal agar (OMA), the WT strain and the ectopic strain developed conidiation colonies (Figure 2A; OMA, WT; arrowheads). In contrast, Δ*rho4* strains were strongly impaired in vegetative growth. While the WT and the ectopic strain showed an increase in colony diameter of approximately 10 mm per day on OMA, the colonies of the Δ*rho4* strains expanded by only 1 mm per day (Figure 2B). Hyperpigmentation and reduced radial growth rates are indicative of cell wall defects, as shown in previous studies with cell wall mutants of *C. graminicola* [5,16,29]. These preceding studies have also shown that growth rates are partially restored by supplementing growth media with osmolytes. On OMA containing either 0.15 M KCl or 1 M sorbitol (Figure 2A; OMA + KCl or OMA + sorbitol), colony diameters of Δ*rho4* mutants increased, as compared to growth in the absence of osmolytes, supporting the hypothesis of cell wall defects in Δ*rho4* strains. As in other cell wall mutants of *C. graminicola*, colonies of Δ*rho4* strains showed dark brownish pigmentation of the mycelium and underlying substratum (Figure 2A).

Cell wall defects in Δ*rho4* mutant strains were further confirmed by differential interference contrast (DIC), bright field (BF), and fluorescence microscopy of Calcofluor White (CFW)-stained hyphae (Figure 2C). The WT strain developed filamentous hyphae with regularly spaced septae (Figure 2C, WT CFW; arrowheads). In contrast, Δ*rho4* strains showed hyphal swellings differing in size and abundance, reminiscent of those observed in vegetative hyphae of *GLS1*-, *KRE5*-, *KRE6*-, *GPI8*-, *GPI12*-, and *GAA1*-RNAi strains defective in β-glucan or glycosylphosphatidylinositol (GPI) anchor formation [5,16,43] (Figure 2C, Δ*rho4* DIC). BF microscopy revealed intrahyphal hyphae (Figure 2B, Δ*rho4* BF; asterisk) within the swellings (Figure 2B, Δ*rho4* BF; arrowhead). Interestingly, Calcofluor White staining of cell wall chitin uncovered compromised septum formation in Δ*rho4* mutants (Figure 2C; compare WT CFW and Δ*rho4* CFW, arrowheads). Moreover, Δ*rho4* cell wall swellings fluoresced more strongly than non-swollen regions (Figure 2C; Δ*rho4* CFW, asterisk) and the cell walls of the WT strain, possibly due to increased compensatory chitin contents.

Comparison of mycelial growth of the WT and Δ*rho4* strains on different antifungal chemistries revealed that colony development of the Δ*rho4* strain was hardly affected by the Echinocandin-based β-1,3-glucan synthase inhibitor Caspofungin [44,45]. The fact that Caspofungin strongly inhibited growth of the WT, but did not further reduce growth of the Δ*rho4* strains (Figure 2D,E), suggests a role of Rho4 in the activation of Gls1 and β-1,3-glucan synthesis. Comprehensibly, in the absence of Rho4, Gls1 would show no or low activity, and the addition of a β-1,3-glucan synthesis inhibitor would not be able to promote this effect further.

Synthesis of β-1,3-glucan is indispensable for asexual sporulation [5]. To investigate the role of Rho4 in conidiation, both strains were grown on potato dextrose agar (PDA). The WT strain started conidiation at 7 days post-inoculation (dpi) (Figure 3A; WT, 7 dpi), and exhibited large numbers of orange-colored acervuli at 14 dpi (Figure 3A; WT, 14 dpi; arrows). By contrast, the Δ*rho4* strain formed colonies with pronounced aerial hyphae without visible acervuli (Figure 3A; Δ*rho4*, 14 dpi). Microscopy of plate washing fluids confirmed that Δ*rho4* had severely reduced conidiation competence, as compared with the WT strain (Figure 3B). Conidia formed by Δ*rho4* had an altered shape and significantly reduced length (Figure 3C,D).

To investigate the role of Rho4 in β-1,3-glucan formation in more detail, we performed ratio imaging microscopy, employing fluorescent Aniline Blue and red-fluorescing Wheat Germ Agglutinin (WGA)-Alexa Fluor 647 conjugate for β-1,3-glucan and chitin staining, respectively (Figure 4). Vegetative hyphae of the *C. graminicola* WT strain were brightly stained by Aniline Blue. In contrast, Δ*rho4* mutant cells were not stained by fluorescent Aniline Blue fluorochrome (Figure 4A,B; β-1,3-glucan, arrowheads), supporting the idea that Rho4 is required for post-translational activation of the β-1,3-glucan synthase protein Gls1 and for β-1,3-glucan synthesis (see Figure 2D,E).

In contrast, chitin staining yielded negligible red fluorescence in the WT, but strongly fluorescing swellings of hyphae of the Δ*rho4* strain (Figure 4A,B; chitin). Thus, contrary to vegetative hyphae of the WT strain, which exhibited a high β-1,3-glucan-to-chitin fluorescence ratio, this ratio was low in hyphae of Δ*rho4* strains (Figure 4C). In accordance with the observation that chitin contents were massively increased in *C. graminicola* Δ*rho4* strains, Caspofungin-treated *A. fumigatus* and *C. albicans* strains exhibited strongly increased cell wall chitin contents [46,47]. Increased chitin staining in hyphal cell wall swellings of Δ*rho4* strains of *C. graminicola* (Figure 4A; Δ*rho4*, arrows) may correspondingly indicate compensation of cell wall defects by an increased incorporation of chitin into the cell walls of Δ*rho4* strains.

Collectively, these data highlight a key role of Rho4 of *C. graminicola* in β-1,3-glucan synthesis and cell wall integrity in vegetative hyphae and conidia, and suggest that other Rho GTPases of this fungus are unable to complement cell wall defects caused by the deletion of *RHO4*.

### 3.3. RHO4 Is Required for Invasive Growth, Infection Structure Differentiation, and Full Virulence

Apical growth and the ability to form infection hyphae with rigid cell walls are pre-requisites for plant invasion and pathogenicity [48]. To compare invasive growth rates of the WT, Δ*rho4*, and ectopic strains of *C. graminicola*, we employed race tubes (Figure 5A) [49] containing PDA supplemented with 0.15 M KCl and increasing agar concentrations. The highest invasive growth rates of the WT and ectopic strains were measured at agar concentrations of 1–2% (*w*/*v*), and hyphae of the WT and ectopic strains were even able to invade PDA containing 4% (*w*/*v*) agar, although at low growth rates (Figure 5B). By contrast, Δ*rho4* mutants were unable to invade agar at any of the concentrations tested (Figure 5B), suggesting that structural cell wall defects interfered with the invasion of solid substrata.

In order to test the ability of Δ*rho4* mutants to differentiate infection structures and to invade the leaves of the host plant, leaves of the third youngest fully expanded leaf of 2-week-old maize (*Z. mays* cv. Badischer Landmais) plants were inoculated with a conidial suspension containing 10^3^ conidia per 10 μL. Conidia of the WT strain germinated and differentiated strongly melanized appressoria (Figure 6A). Hyphopodia were rarely observed (Figure 6G). In contrast, the Δ*rho4* mutant showed significant rates of bipolar germination (Figure 6C; arrows; Figure 6E), and long hyphae with hyphopodia were frequently found, resulting in a high hyphopodia/appressoria ratio (Figure 6C,F,G). On the leaf surface, the Δ*rho4* mutant formed extended hyphal coils (Figure 6C; dashed box). Further supporting the idea that the Δ*rho4* mutant had defective cell walls, hyphopodia of this mutant germinated laterally, often in a repeated manner (Figure 6C; insert). Increased rates of lateral germination had previously been reported for cell wall-defective *GLS1*-RNAi strains [5]. Importantly, the Δ*rho4* mutant exhibited significantly reduced penetration competence, as indicated by the formation of less than 40% infection hyphae at 72 hpi, as compared to ~90% infection hyphae differentiated by the WT strain at that time. While the WT strain formed voluminous biotrophic infection hyphae (Figure 6B; ih), infection hyphae formed by the Δ*rho4* strain appeared fringed and distorted (Figure 6D; ih). Comparably, on the alternative host, *Allium cepa* cv. Shakespeare, the WT strain formed appressoria, which invaded the onion cell and differentiated smooth infection hyphae (Appendix A). As on maize, the Δ*rho4* mutant formed long hyphae on the onion surface and large numbers of hyphopodia, which germinated laterally (Appendix A; arrowheads; and Appendix A). Host cell wall penetration and the formation of infection hyphae were significantly reduced in the Δ*rho4* mutant (Appendix A). Intriguingly, in onion epidermal cells the Δ*rho4* mutant formed infection hyphae at a high density (Appendix A). These hyphae showed large protrusions (Appendix A; arrowheads), again, indicative of hyphal cell wall defects.

Compromised differentiation of infection cells on the host epidermis, reduced penetration rates, and formation of distorted infection hyphae in the host epidermal cell suggested that the Δ*rho4* mutant may exhibit clearly reduced virulence. To test this hypothesis, we inoculated intact and wounded segments of the third youngest fully expanded leaf of two-weeks-old maize (*Z. mays* cv. Badischer Landmais) plants with 10 μL droplets containing 10^6^ conidia/mL 0.02% (*w*/*v*) Tween 20. At 4 dpi, the WT strain had caused clear disease symptoms on both non-wounded and wounded leaves, with, as expected, stronger disease symptoms on wounded leaves (Figure 7A; WT). In contrast, on intact leaf segments, the Δ*rho4* mutants caused only minor disease symptoms (Figure 7A; Δ*rho4*, non-wounded). On wounded leaves, allowing the pathogen to invade the leaf without differentiation of appressoria or hyphopodia, the WT strain caused severe and stronger anthracnose symptoms than on intact leaves; anthracnose symptoms caused by Δ*rho4* mutants on wounded leaves were still minor (Figure 7A; WT, Δ*rho4*, wounded). The macroscopically observed intensity of disease symptom severities was confirmed by qPCR analyses performed with primers targeting the ITS2 of the rDNA cluster [30] (Figure 7B).

In summary, the evaluation of disease symptoms, microscopy, quantification of infection structure differentiation, and qPCR analyses showed that the Rho4 protein characterized here is a novel virulence factor of the maize anthracnose fungus *C. graminicola*.

## 4. Discussion

Yeasts and filamentous fungi harbor a large repertoire of genes encoding Ras superfamily GTPases, including Rho GTPases. Rho proteins are important regulators of several developmental aspects such as cell polarity, vesicle trafficking, and cell wall biogenesis. A recent meta-study used the genomes of 56 fungal species belonging to eight distinct fungal phyla to identify and compare Ras, Rho, Rab, and Ran family members, with the maize anthracnose pathogen *C. graminicola* also included in this study [19]. While the model yeasts *S. cerevisiae* and *S. pombe* contain five Rho proteins, only homologs of Rho1p–Rho4p have been identified in the vast majority of filamentous fungi [19]. In *S. cerevisiae* and *S. pombe*, Rho1p physically interacts with and activates β-1,3-glucan synthase [20,23,26,50]. Thus, not surprisingly, Δ*rho1* cells of yeast exhibit a growth arrest, and cells lyse due to severe cell wall defects [51]. Not only in yeasts, but also in filamentous fungi, have Rho1 GTPases been identified as activators of β-1,3-glucan synthases. By generating conditional mutants of the essential *RHO1* gene of *N. crassa*, Seller and co-workers showed that Rho1 functions as a regulatory subunit of β-1,3-glucan synthase required for hyphal integrity and polarity [52]. Additionally, in different *Aspergillus* spp., Rho1 is a viability factor and acts as a subunit of the β-1,3-glucan synthase complex [24,25,53,54,55]. Taken together, a large body of literature indicates that Rho1 is required for β-1,3-glucan synthesis, cell wall integrity, and viability in many fungi [19,22,56].

Intriguingly, we identified a Rho4-like GTPase, the deletion of which caused strong growth and hyphal defects reminiscent of those observed in *GLS1*-RNAi strains [5]. The annotation of the small Rho GTPase as *C. graminicola* Rho4, though previously miss-annotated as a Rho1 protein [57], is in accordance with the protein size, G-box positions, and amino acid sequence of other Rho4 GTPases. The new annotation is also in agreement with the annotation of this protein published elsewhere [19,42].

The striking similarity of the cell wall defects displayed by the *C. graminicola* Δ*rho4* mutants and *GLS1*-RNAi strains [5] suggests that both genes may serve the same cell wall function. Recently, a double tagging plasmid system was employed to test the physical interaction of the *C. graminicola* β-1,3-glucan synthase Gls1 with Rho4, formerly annotated as Rho1. Gls1 fused to the *N*-terminus and Rho4 fused to the *C*-terminus of split EYFP yielded strong fluorescence in bimolecular fluorescence complementation (BiFC) experiments [57]. Thus, the phenotypic similarity of *GLS1*-RNAi strains and Δ*rho4* deletion mutants as well as the BiFC experiment strongly suggest a close spatial and functional association of Gls1 and Rho4 in the maize anthracnose fungus and a role of Rho4 in the activation of β-1,3-glucan synthase. In line with these data, ratio-imaging microscopy revealed significantly reduced β-1,3-glucan contents in cell walls of vegetative hyphae of Δ*rho4* mutants. Contrarily, chitin contents were clearly increased, which may be interpreted as a compensatory response to rescue cell wall defects caused by insufficient amounts of β-1,3-glucan. Interestingly, increased chitin contents have been observed as a compensatory response to reduced β-1,3-glucan contents in hyphal walls in *A. fumigatus* and *C. albicans* strains after treatment with the β-1,3-glucan synthase inhibitor Caspofungin [46,47]. Thus, decreased β-1,3-glucan and increased chitin contents observed by ratio imaging in Δ*rho4* mutants further support a direct role of *C. graminicola* Rho4 in β-1,3-glucan synthesis. Remarkably, also in the head blight fungus of wheat, *Fusarium graminearum*, staining of conidia and hyphae with the chitin dye Calcofluor White yielded stronger fluorescence in Δ*rho4* mutants than in the WT strain. Similar to *C. graminicola* Δ*rho4* mutants, the corresponding mutants of *F. graminearum* and *A. niger* showed drastically reduced growth and conidiation rates [28,54]. Furthermore, in the *C. graminicola* Δ*rho4* mutant, septum formation appeared to be compromised. In agreement with these findings, in the closely related fungi *N. crassa and A. nidulans*, Rho4 was identified as a dynamic component of the contractile actin ring (CAR) required for septum formation, with Rho4 likely representing an activator of formins at septation sites [58,59]. In the fission yeast, *S. pombe*, both *rho4*^+^ deletion, as well as *rho4*^+^ overexpression, caused strong cell wall defects, suggesting a prominent function of this GTPase in cell wall integrity [60]. Although an important role of Rho4 in cell wall biogenesis has been reported in *A. niger* and *A. fumigatus* [54,55], a direct physical interaction with and activation of β-1,3-glucan synthase has only been reported in *C. graminicola* so far [57]. The fact that intrahyphal hyphae have been noticed in *KRE5*- and in *KRE6*-RNAi strains of *C. graminicola* may indicate that Rho4 plays a role in β-1,6-glucan synthesis [16]. Alternatively, a shift in the β-1,3-to-β-1,6-glucan-to-chitin ratio may lead to cell wall disorders, as visualized by the occurrence of intrahyphal hyphae. In this context, it should be mentioned that intrahyphal hyphae characteristically occur in mutants defective in chitin synthases with a myosin-like motor domain [16,29], and references therein).

As the synthesis of structural cell wall polymers is indispensable for fungal virulence [6], it is not surprising that β-1,3-glucan synthase-activating Rho GTPases are required in plant infection. For example, Δ*rho3* mutants of the rice blast fungus *M. oryzae* differentiate appressoria with reduced turgor pressure, reduced penetration, and reduced infection hypha differentiation rates. As a result, disease symptoms hardly developed on rice leaves. Interestingly, overexpression of *RHO3* enhanced virulence. On the alternative host onion, microscopy revealed that infection hyphae formed by Δ*rho3* appressoria were able to invade epidermal cells and were morphologically normal [61]. In the rice blast fungus, two additional Rho proteins, i.e., a Cdc42- and a Rac1-homolog were also required for full virulence on rice [62,63]. In the wheat head blight fungus *F. graminearum*, the entire repertoire of Rho GTPases has been functionally characterized [28]. Single deletion mutants of all genes showed impaired growth and conidial irregularities. Rho2 and Rho4 were thought to be involved in cell wall integrity, as deduced from inhibition of growth on a medium containing the cell wall perturbing agent Calcofluor White. However, unlike Δ*rho4* mutants of *C. graminicola*, microscopy did not show hyphal distortions such as swellings in the corresponding mutants of the *Fusarium* head blight fungus. Interestingly, inoculation of single wheat kernels revealed that the Δ*rho4* strain of *F. graminearum* was unable to produce head blight symptoms, indicating a role of *RHO4* in pathogenicity [28]. This finding was indirectly supported by a Δ*bud3* deletion mutant of this fungus. In the yeast two-hybrid assays, the guanine nucleotide exchange factor Bud3 strongly interacted with both the GDP- as well as the GTP-bound state of Rho4, but only weakly with Rho2, Rho3, and Cdc42. Interestingly, the Δ*bud3* deletion mutant, like the Δ*rho4* mutant of *F. graminearum*, was unable to cause head blight disease symptoms on wheat [25].

Studies of the Rho GTPase-coding genes of the plant pathogens discussed above indicate that different Rho GTPases contribute to cell wall biogenesis, infection structure differentiation, and virulence in distinct fungi. In addition to distinct functions of Rho proteins with partial functional overlaps [19,52,56], job sharing between distinct Rho proteins may be required in some processes. The ability of another Rho protein to complement Rho4 at a low level may explain why in *C. graminicola* deletion of *GLS1* is lethal [5], but deletion of *RHO4* is not.

In summary, we demonstrated that the small Rho GTPase-coding gene *RHO4* of *C. graminicola* is required for β-1,3-glucan synthesis, cell wall integrity, growth of vegetative hyphae and conidiation, infection structure differentiation, and full virulence. As the activation of β-1,3-glucan synthases has previously been attributed to Rho1 proteins, our findings strengthen the concept of distinct but partially overlapping functions of small Rho GPTases in fungi.

## Figures and Tables

**Figure 1 jof-08-00997-f001:**
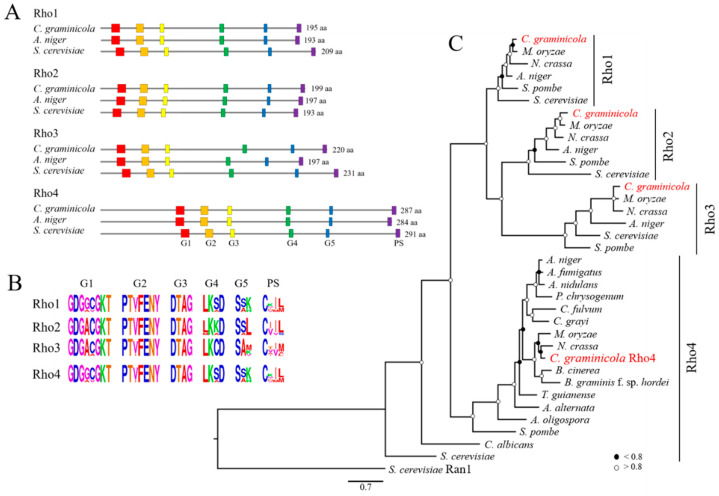
Phylogenetic tree and organization of conserved domains in Rho proteins of *C. graminicola*, as compared with Rho GTPase proteins of yeasts and other filamentous fungi. (**A**) Size and positions of G-boxes and prenylation sites of Rho proteins of *C. graminicola* and the filamentous and yeast model fungi *A. niger* and *S. cerevisiae*. Amino acid (aa) numbers of the Rho proteins are given on the right side. (**B**) Sequence logo indicating amino acid sequence conservation of G-boxes and prenylation sites. Comparisons were made using the amino acid sequences of *C. graminicola*, of the model filamentous ascomycetes *Magnaporthe oryzae*, *N. crassa*, and *Aspergillus niger*, and of the yeasts *S. cerevisiae* and *S. pombe*. (**C**) Rooted phylogenetic tree indicates close relatedness of Rho1–Rho4 proteins of different filamentous ascomycetes and yeasts. The *S. cerevisiae* Ran1 protein served as an outgroup. Solid and open circles indicate bootstraps higher or lower than 80.

**Figure 2 jof-08-00997-f002:**
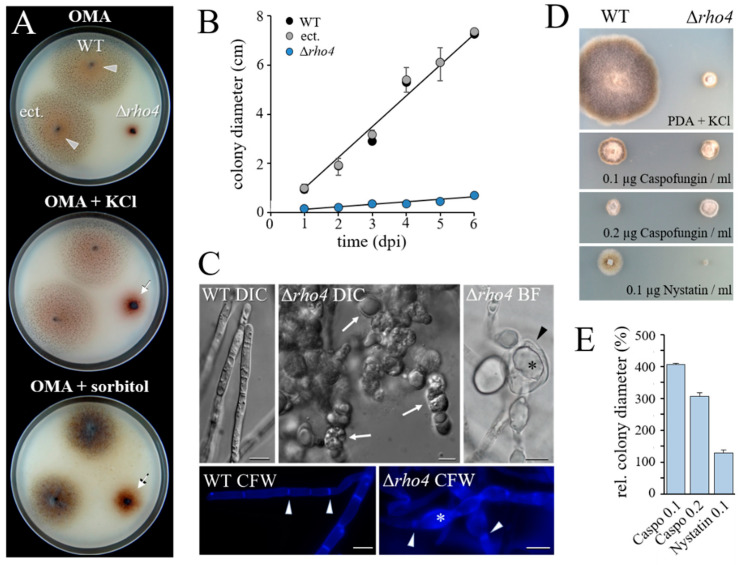
Relevance of *C. graminicola* Rho4 for growth, hyphal morphology, and sensitivity to Caspofungin. (**A**) Growth of the WT, Δ*rho4*, and ectopic strains on OMA or OMA supplemented with osmolytes (OMA + KCl or OMA + sorbitol). Arrowheads point at salmon-colored acervuli with conidia. Arrows indicate the hyper-pigmented Δ*rho4* strain. Plates were photographed at 4 dpi. (**B**) Growth of WT, ectopic, and Δ*rho4* strains on OMA. Error bars indicate SDs. (**C**) Microscopy indicated severe cell wall defects of Δ*rho4* strains. As compared with the WT strain, DIC microscopy of the Δ*rho4* strain displayed massive ballooning of vegetative hyphae (Δ*rho4* DIC, arrows; compare with WT DIC). Bright-field microscopy showed intrahyphal hyphae (Δ*rho4* BF, asterisk) developing within hyphae (Δ*rho4* BF, arrowhead). While Calcofluor White staining of WT hyphae revealed clear septation (WT CFW, arrowheads), brightly stained swellings (Δ*rho4* CFW asterisk) and diffuse septa (Δ*rho4* CFW arrowheads) were seen in hyphae of the mutant. Microscopy was done at 4 dpi. Scale bars are 10 μm. (**D**) Colony growth of WT and Δ*rho4* strains on PDA + KCl, and on PDA + KCl amended with the antifungals Caspofungin or Nystatin. Growth of the WT strain is strongly retarded by all antifungals. The Δ*rho4* strain is not inhibited by Caspofungin, but growth is completely blocked by Nystatin. The size of the agar block used as a Δ*rho4* inoculum is clearly visible. Plates were photographed at 7 dpi. (**E**) Growth inhibition ratio of Δ*rho4* vs. WT strain. Growth on media containing antifungals is expressed relative to growth on PDA + KCl. Error bars indicate SDs.

**Figure 3 jof-08-00997-f003:**
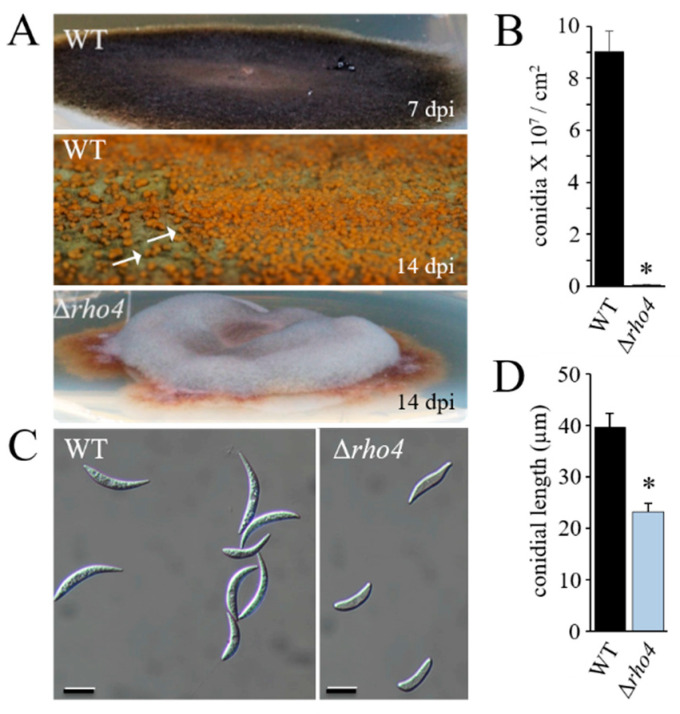
Asexual sporulation of the *C. graminicola* WT and Δ*rho4* strains on PDA amended with KCl. (**A**) After 7 dpi the WT had formed a dark mycelium, which developed orange-colored acervuli (arrows) by 14 dpi. The Δ*rho4* strain developed aerial mycelium without visible acervuli. (**B**) Quantification of conidia formed by the WT and Δ*rho4* strains. (**C**) Shape of conidia formed by the WT and Δ*rho4* strains. Scale bars are 20 μm. (**D**) Length of conidia formed by the WT and Δ*rho4* strains. Error bars in B and D are SDs. Asterisks indicate statistically significant differences (*p* ≤ 0.05).

**Figure 4 jof-08-00997-f004:**
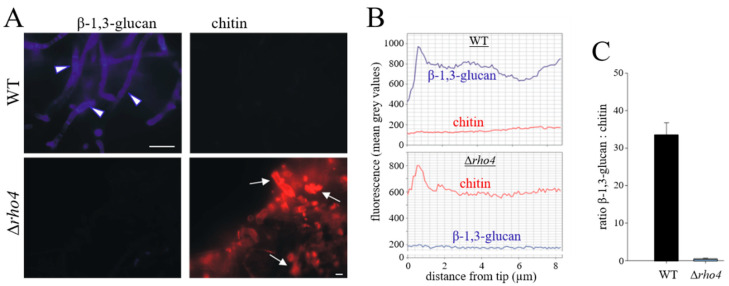
Chitin and β-1,3-glucan abundance in vegetative hyphae of the WT and Δ*rho4* mutant. (**A**) WT strain and Δ*rho4* hyphae were stained with Aniline Blue fluorochrome and Alexa Fluor 647-Wheat Germ Agglutinin conjugate to detect β-1,3-glucan and chitin, respectively. While hyphae of the WT strain showed strong β-1,3-glucan labeling (arrowheads), those of Δ*rho4* strains did not. By contrast, chitin was hardly observed in the WT strain, but abundant in Δ*rho4* hyphae, with prominent labeling at hyphal protrusions (arrows). Scale bars are 10 μm. (**B**) Quantification of fluorescence intensity in hyphae labelled as in (**A**), from the hyphal tip to subapical regions. Virtual line scans confirm strong β-1,3-glucan labeling of the WT and strong chitin labeling of Δ*rho4* hyphae. (**C**) The β-1,3-glucan-to-chitin fluorescence ratios of the WT and Δ*rho4* strains show that the proportion of β-glucan is substantially reduced in Δ*rho4* strains. Error bars indicate SDs.

**Figure 5 jof-08-00997-f005:**
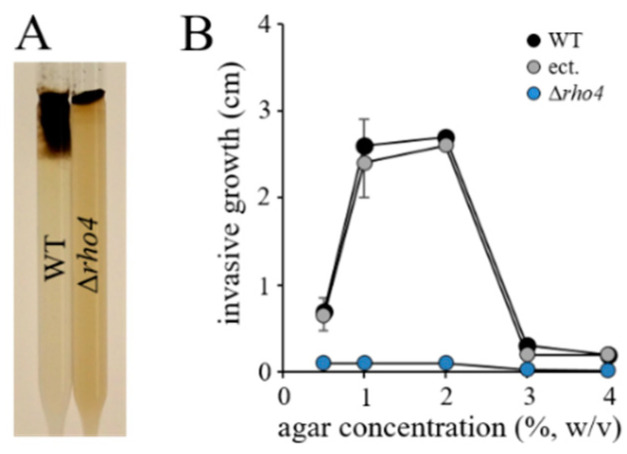
Invasive growth of the WT and strains with homologous and ectopic integration of the deletion cassette in race tubes. (**A**) Race tubes filled with PDA osmotically stabilized with KCl visualized invasive growth of the WT and Δ*rho4* strains. (**B**) Quantification of invasive growth as dependent of the agar concentration. Error bars are SDs.

**Figure 6 jof-08-00997-f006:**
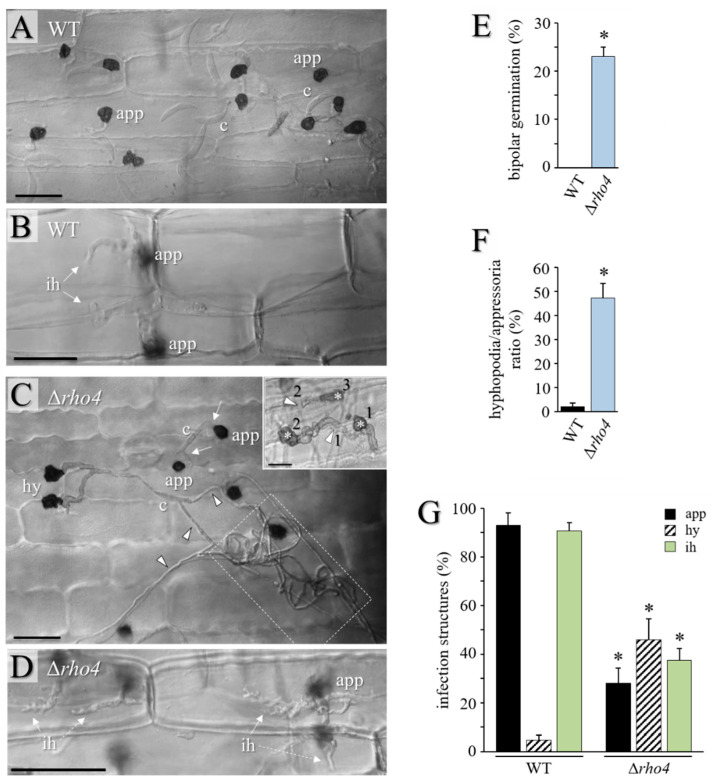
Δ*rho4* strains are defective in infection structure differentiation on maize leaves. (**A**–**D**) DIC microscopy of the infection process of *C. graminicola* WT and Δ*rho4* strains. Conidia (**A**,**C**) of the WT strain germinate and differentiate melanized appressoria ((**A**), app), which invade the epidermal host cell and differentiate infection hyphae ((**B**), ih). The Δ*rho4* mutant, by contrast, exhibits bipolar germination ((**C**), arrows), which form long ((**C**), arrowheads) and often coiled hyphae ((**C**), dashed box) before they differentiate hyphopodia ((**C**), hy). Several hyphopodia of the Δ*rho4* mutant germinate laterally ((**C**), insert; asterisks and numbers 1–3 indicate first–third hyphopodium; arrowheads with numbers 1 and 2 indicate first and second lateral germ tube). Not only hyphopodia, but also appressoria, are formed ((**C**), app). After penetration, the Δ*rho4* mutant forms fringy infection hyphae with numerous small branches ((**D**), ih). Scale bars in (**A**–**C**) are 20 μm; scale bar in (**D**) is 50 μm. (**E**) Rate of bipolar germination of WT and Δ*rho4* conidia. Error bars are SDs; asterisk indicates statistically significant differences (*t*-test, *p* ≤ 0.05). (**F**) Ratio of hyphopodia to appressoria formed by WT and Δ*rho4* strains. Error bars are SDs; asterisk indicates statistically significant differences (*t*-test, *p* ≤ 0.05). (**G**) Appressorium, hyphopodium, and infection hypha differentiation by WT and Δ*rho4* strains. Error bars are SDs; asterisks indicate statistically significant differences of corresponding structures (*t*-test, *p* ≤ 0.05).

**Figure 7 jof-08-00997-f007:**
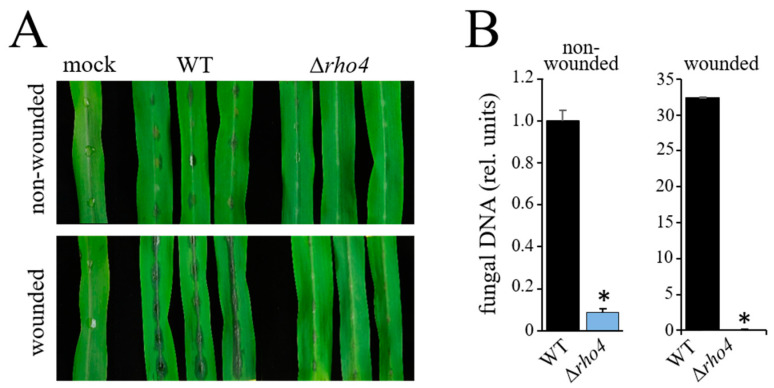
Δrho4 strains are nonpathogenic on wounded and non-wounded leaves. (**A**) Disease symptoms on non-wounded and wounded maize leaves after inoculation with the WT and Δ*rho4* strains. Mock-inoculated leaves were treated with 0.01% (*v*/*v*) Tween 20 and served as controls (mock). Photographs were taken at 4 dpi. (**B**) Quantification of fungal development on non-wounded and wounded maize leaves by qPCR. Three independent biological and four technical replicates were analyzed for WT and Δ*rho4* strains. Error bars are SDs. Asterisks indicate statistically significant differences (*t*-test, *p* ≤ 0.05).

## Data Availability

Not applicable.

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
