# Peer review of "The Small Ras Superfamily GTPase Rho4 of the Maize Anthracnose Fungus Colletotrichum graminicola Is Required for β-1,3-glucan Synthesis, Cell Wall Integrity, and Full Virulence"

_jof, 2022, doi:10.3390/jof8100997_

Round 1
Reviewer 1 Report
This manuscript was well presented which reported the virulence role of GTPase Rho4 in Colletotrichum graminicola, I have fewer suggestions for revision.
The title seems not so interesting, please revise it.
Some of the references were old, please update and replace with most the recent related publications.
Author Response
We thank the reviewers for critically reading and commenting on our manuscript. We fully agree to most of the points raised by the reviewers and have changed the text accordingly.
Reviewer 1:
Comments and Suggestions for Authors
This manuscript was well presented which reported the virulence role of GTPase Rho4 in Colletotrichum graminicola, I have fewer suggestions for revision.
The title seems not so interesting, please revise it.
Response: The title reflects and highlights the most relevant data, at least from the perspective of plant pathologists interested in understanding mechanisms of fungal pathogenicity. We do agree, however, that the title may be a little complex, and we tried to simplify it and eliminated the word ‘hyphal’, as in filamentous fungi cell walls are per se associated with hyphae. However, in case reviewer 1 would suggest an alternative title, we would consider using it.
Some of the references were old, please update and replace with most the recent related publications.
Response: To the best of our knowledge, we have tried to highlight the first discoveries made in the field, and also included follow-up papers if relevant and if they contribute novel aspects to understanding Rho function or cell wall biogenesis. In case we should have missed relevant work, would be grateful if the reviewer would specify this. We would then be glad to include these references.
Reviewer 2:
Comments and Suggestions for Authors
I do find this work interesting and valuable. The manuscript is well written. Some suggestions (mostly editorial) for improving are below:
Line 141, the full name of dpi should be added;
Response: This has been done.
Line 244, 248, "C. graminicola" should be typed in italics;
Response: We have checked the entire manuscript, and in all cases C. graminicola is written in italics now.
Line 249, "RHO4" should be typed in italics, please check the full text;
Response: We have checked the text carefully and think that all gene and protein names are formally correct now. Names of genes are given in italics and capital letters/numbers, whereas proteins are given non-italicized and with the first letter only as a capital letter. Mutants are indicated by a delta, followed by the name of the gene in small letters and in italics. This is in full agreement with gene and protein names in filamentous fungi.
Line 258, Magnaporthe oryzae, N. crassa, and Aspergillus niger, the name of strains should be typed in italics, please check the full text.
Response: As with the previous comment, we have checked the manuscript again, and all species names are given in italics.
Line 282, Δrho4 deletion strains, rho4 should be typed in italics, please check the full text;
Response: Δrho4 is now Δrho4.
Figure 7 is disordered.
Response: This is indeed irritating. In our PDF version, disordering does not occur, and we hope that in the revised version, Fig. 7 is technically acceptable.
The 1st paragraph in Discussion should be simplified
Response: Indeed, some sentences were too complex. We have simplified these and hope the first paragraph of the Discussion section is now easier to read.

Reviewer 2 Report
I do find this work interesting and valuable. The manuscript is well written. Some suggestions (mostly editorial) for improving are below:
Line 141, the full name of dpi should be added;
Line 244, 248, "C. graminicola" should be typed in italics;
Line 249, "RHO4" should be typed in italics, please check the full text;
Line 258, Magnaporthe oryzae, N. crassa, and Aspergillus niger, the name of strains should be typed in italics, please check the full text.
Line 282, Δrho4 deletion strains, rho4 should be typed in italics, please check the full text;
Figure 7 is disordered.
The 1st paragraph in Discussion should be simplified.
Author Response
Reviewer 2:
Comments and Suggestions for Authors
I do find this work interesting and valuable. The manuscript is well written. Some suggestions (mostly editorial) for improving are below:
Line 141, the full name of dpi should be added;
Response: This has been done.
Line 244, 248, "C. graminicola" should be typed in italics;
Response: We have checked the entire manuscript, and in all cases C. graminicola is written in italics now.
Line 249, "RHO4" should be typed in italics, please check the full text;
Response: We have checked the text carefully and think that all gene and protein names are formally correct now. Names of genes are given in italics and capital letters/numbers, whereas proteins are given non-italicized and with the first letter only as a capital letter. Mutants are indicated by a delta, followed by the name of the gene in small letters and in italics. This is in full agreement with gene and protein names in filamentous fungi.
Line 258, Magnaporthe oryzae, N. crassa, and Aspergillus niger, the name of strains should be typed in italics, please check the full text.
Response: As with the previous comment, we have checked the manuscript again, and all species names are given in italics.
Line 282, Δrho4 deletion strains, rho4 should be typed in italics, please check the full text;
Response: Δrho4 is now Δrho4.
Figure 7 is disordered.
Response: This is indeed irritating. In our PDF version, disordering does not occur, and we hope that in the revised version, Fig. 7 is technically acceptable.
The 1st paragraph in Discussion should be simplified
Response: Indeed, some sentences were too complex. We have simplified these and hope the first paragraph of the Discussion section is now easier to read.